# Chronic Administration of Diethylnitrosamine and 2-Acetylaminofluorene Induces Hepatocellular Carcinoma in Wistar Rats

**DOI:** 10.3390/ijms24098387

**Published:** 2023-05-07

**Authors:** Jaime Sánchez-Meza, Marina Campos-Valdez, José Alfredo Domínguez-Rosales, Juliana Marisol Godínez-Rubí, Sarai Citlalic Rodríguez-Reyes, Erika Martínez-López, Guillermo M. Zúñiga-González, Laura Verónica Sánchez-Orozco

**Affiliations:** 1Instituto de Enfermedades Crónico Degenerativas, Centro Universitario de Ciencias de la Salud, Universidad de Guadalajara, Guadalajara 44340, Mexico; jaimesanchez96@hotmail.com (J.S.-M.); campos.ibt@gmail.com (M.C.-V.); dominque14@yahoo.com (J.A.D.-R.); 2Laboratorio de Patología Diagnóstica e Inmunohistoquímica, Departamento de Microbiología y Patología, Centro Universitario de Ciencias de la Salud, Universidad de Guadalajara, Guadalajara 44340, Mexico; juliana.godinez@academicos.udg.mx; 3Instituto de Nutrigenética y Nutrigenómica Traslacional, Centro Universitario de Ciencias de la Salud, Guadalajara 44340, Mexicoerika.martinez@academicos.udg.mx (E.M.-L.); 4Laboratorio de Mutagénesis, Centro de Investigación Biomédica de Occidente, Instituto Mexicano del Seguro Social, Guadalajara 44340, Mexico; mutagenesis95@hotmail.com

**Keywords:** diethylnitrosamine, 2-acetylaminofluorene, hepatocellular carcinoma, Wistar rats, *TGF-β*, alpha-fetoprotein

## Abstract

This study aimed to analyze the biochemical, histological, and gene expression alterations produced in a hepatocarcinogenesis model induced by the chronic administration of diethylnitrosamine (DEN) and 2-acetylaminofluorene (2-AAF) in Wistar rats. Thirteen rats weighing 180 to 200 g were divided into two groups: control and treated. Rats in the treated group were administered an intraperitoneal (i.p.) injection of DEN (50 mg/kg/week) and an intragastric (i.g.) dose of 2-AAF (25 mg/kg/week) for 18 weeks. The treated group had significant increases in their total cholesterol, HDL-C, AST, ALT, ALKP, and GGT levels. Furthermore, a histological analysis showed the loss of normal liver architecture with nuclear pleomorphism in the hepatocytes, atypical mitosis, and fibrous septa that were distributed between the portal triads and collagen fibers through the hepatic sinusoids. The gene expressions of 24 genes related to fibrosis, inflammation, apoptosis, cell growth, angiogenesis, lipid metabolism, and alpha-fetoprotein (*AFP*) were analyzed; only *TGFβ*, *COL1α1*, *CYP2E1*, *CAT, SOD*, *IL6*, *TNF-α*, and *ALB* showed significant differences when both groups were compared. Additionally, lung histopathological alterations were found in the treated group, suggesting metastasis. In this model, the chronic administration of DEN+2-AAF induces characteristic alterations of hepatocellular carcinoma in Wistar rats without *AFP* gene expression changes, highlighting different signatures in hepatocellular carcinoma heterogeneity.

## 1. Introduction

Hepatocellular carcinoma (HCC) represents the main primary liver carcinoma in which adult hepatocytes have been proposed to transform directly into HCC cells [1,2]. Patients with chronic liver disease have sustained hepatic inflammation, fibrosis, oxidative stress disequilibrium, and aberrant hepatocyte regeneration; these anomalies can cause cirrhosis that culminate in the formation of dysplastic nodules leading to preneoplastic lesions [3,4,5]. HCC, as a type of cancer, is the sixth in terms of incidence, and the third cause of cancer-related mortality worldwide [6]. In 2020, there were more than 900,000 new cases of HCC and 830,180 deaths caused by this disease [6]. The main risk factors for HCC development are the hepatitis B or C virus (HBV or HCV), chronic infection, and alcohol abuse. Nonetheless, aflatoxin B1 exposition, non-alcoholic fatty liver disease (NAFLD), as well as environmental and industrial toxic chemicals are also associated with this cancer [3,4,5]. Risk factors associated with NAFLD, such as obesity, diabetes, insulin resistance, and the metabolic syndrome, have also been independently associated with HCC development [7,8]. Hepatic lipid accumulation itself is a common complication from these pathologies and can generate carcinogenic conditions such as lipotoxicity, including cholesterol-related lipotoxicity, and oxidative DNA damage. These phenomena could explain the association, particularly in the absence of cirrhosis, between NAFLD and HCC [8]. In addition, the high prevalence of obesity worldwide and the unavoidable exposure to toxic compounds could prognosticate a serious outlook for worldwide HCC incidence.

If public health campaigns promoting healthy lifestyles do not result in an effective impact and most people continue with obesogenic diets that predispose people to insulin resistance, diabetes, NAFLD, or metabolic syndrome, HCC could be one of the most common causes of death caused by cancer in the near future [7,8]. In addition, the unavoidable exposure of humans to toxic chemical compounds could further complicate the HCC outlook in the real world. Diethylnitrosamine (DEN) is a chemical found in tobacco smoke and food products such as meat and alcoholic beverages [9]. There are reports on the establishment of experimental HCC animal models that were constructed with the administration of DEN as the only agent or were combined with tumor-promoting conditions (such as the administration of phenobarbital, carbon tetrachloride, ethanol, or high-fat diets). Different hepatocarcinogenic protocols using these substances have been reported, with variations in dosage, time points of administration, and protocol time lapse [10]. Furthermore, 2-acetylaminoflourene (2-AAF)—developed as an insecticide for agricultural use—was discontinued due to the discovery of its carcinogenic and mutagenic properties. Currently, its use is limited to experimental models in the study of DNA adduct structures, DNA repair mechanisms, carcinogenesis, and mutagenesis [11]. The carcinogenic and mutagenic effects of 2-AAF are derived from its alteration of signaling pathways at an epigenetic level, which promote the incontrollable proliferation of damaged cells [12].

DEN and 2-AAF are biotransformed, mainly by cytochrome P450 2E1 (CYP2E1), in the liver by the cytochrome P450 enzymatic system. As a consequence, the generated metabolites interact with DNA to form N7-methylguanine, O^6^-ethylguanine, and N-(deoxyguanosin-8-yl)-2-acetylaminofluor, which are the main DNA adducts associated with the carcinogenic effects of DEN and 2-AAF, respectively. In addition, DEN biotransformation induces an increase in reactive oxygen species (ROS), which can also provoke the formation of DNA adducts and/or even alter the cell signaling pathways involved in tumor progression and/or metastasis [13,14,15,16].

The combined treatment of DEN and 2-AAF in animal models for the development of HCC has been widely used to study the molecular basis of different stages of carcinogenesis, as well as to evaluate different treatment options [17,18]. In addition, 2-AAF acts as a tumor promoter by blocking the proliferation of undamaged hepatocytes (due to its mitogen inhibitory activity) and by simultaneously stimulating the proliferation of DEN-initiated cells; this induces the formation of preneoplastic foci and hyperplastic nodules in the liver. The proliferating hepatic progenitor cells (HPCs) are resistant to the toxicity of this chemical and favor the induction of hepatic tumors [12,19].

Heterogeneity has been identified in human HCC; this heterogeneity is both intratumoral and interpatient. Intratumoral heterogeneity negatively affects the prognosis of patients with HCC because it favors tumor growth, metastasis, recurrence, and drug resistance, which limits the choice of the most effective therapy for each patient, whereas the interpatient heterogeneity is related to personalized medicine. The heterogeneity in HCC involves morphological, genetic, epigenetic, and metabolic aspects, so it is necessary to design animal models that allow for studying HCC from all these perspectives [20]. As an example, the high expression of AFP has been correlated with the MYC and AKT activation signature, whereas HCC with a WNT activation signature and high levels of *TGF-β* were correlated with no changes in AFP and poor survival [21]. Once HCC is established and diagnosed, treatment must be assigned according to the stages of the tumor; generally, for patients in the early stage of HCC (one to three nodules smaller than 3 cm), partial hepatectomy, liver transplantation, or image-directed ablation should be preferably performed. These therapies offer the best results with a survival range of 70% to 80% in 5 years and a perioperative mortality of 3% [22]. Intermediate-stage patients with multilobar disease and adequate liver function are candidates for transarterial chemoembolization; advanced-stage patients, on the other hand, with portal thrombosis or extrahepatic spread, should be treated with systemic therapy [22]. Atezolizumab plus bevacizumab, sorafenib, lenvatinib, regorafenib, cabozantinib, and ramucirumab are systemic therapies that have been approved for the treatment of HCC; a combination of atezolizumab (anti-PDL1 antibody) and bevacizumab (anti-VEGF antibody) was approved for advanced-stage HCC cases as first-line therapies, but if the patient has contraindications to these drugs, sorafenib and lenvatinib can be used. In fact, sorafenib and lenvatinib remain the most effective first-line single-drug therapies [22].

In the study of in vivo models of HCC, it is important to consider that there are differences in the susceptibility to hepatocarcinogenesis among murine strains due to their genetic differences. Furthermore, each protocol for HCC induction generates heterogeneous gene expression changes. Consequently, different signatures in the signaling routes, molecular markers, and metabolism can be observed depending on the chosen strain and HCC induction protocol [23].

The development of different HCC experimental models is a valuable tool through which to understand the diverse mechanisms related to HCC induction, establishment, progression, and metastasis. Furthermore, trying to extrapolate the data obtained from them with what is known about the genotypic and phenotypic variability of this pathology in humans is a key part of developing effective treatments for this cancer. Recently, Castro-Gil et al., in 2021, reported an HCC model induced with the chronic administration (16 weeks) of DEN and 2-AAF in Fisher 344 rats (an HCC-susceptible strain). In this same study, partial hepatectomy was not performed (in contrast to other HCC models) and it was substituted by chronic oral administration of 2-AAF in order to promote and accelerate the hepatocarcinogenesis induced by DEN. In the aforementioned study, an enrichment of HPCs during hepatocarcinogenesis was found, supported by the presence of HPC markers as early as week 6 of treatment [18].

The present study was designed to determine if the chronic administration of DEN and 2-AAF in Wistar rats (a rat strain with intermediate resistance to HCC development) could be an efficient protocol through which to establish a useful in vivo HCC model—which may simulate the pathological process of HCC that originates from exposure to toxic chemicals—and to study the histological and biochemical alterations, as well as the gene expression changes in relevant genes associated with hepatocarcinogenesis.

Hepatocarcinogenesis was induced by a weekly administration of DEN and 2-AAF for 18 weeks in Wistar rats in order to evaluate blood biochemical parameters, histopathologic alterations, and gene expression changes related to the following: fibrosis [transforming growth factor beta 1 (*TGFβ1*), collagen type I alpha 1 chain (*COL1α1*), collagen type III alpha 1 chain (*COL3α1*), and TIMP metallopeptidase inhibitor 1 (*TIMP1*)]; inflammation [interleukin 1 beta (*IL1β*), interleukin 6 (*IL6*), and tumor necrosis factor alpha (*TNFα*)]; oxidative stress [catalase (*CAT*), superoxide dismutase 1 (*SOD*), and nitric oxide synthase (*iNOS*)]; cellular proliferation [proliferating cell nuclear antigen (*PCNA*) and hepatocyte growth factor (*HGF*)]; apoptosis [BCL2-associated X, apoptosis regulator (*BAX*) and BCL2 apoptosis regulator (*BCL-2*)]; angiogenesis [hypoxia inducible factor 1 subunit alpha (*HIF1α*) and vascular endothelial growth factor receptor 2 (*VEGF-R2*)]; metabolism [mitogen-activated protein kinase 1 (*ERK-2*), mitogen-activated protein kinase 14 (*P38*), albumin (*ALB*), peroxisome proliferator activated receptor alpha (*PPARα*), carnitine palmitoyltransferase 1A (*CPT1A*), and cytochrome P450 family 2 subfamily E member 1 (*CYP2E1*)]; and the tumor marker alpha-fetoprotein (*AFP*). Additionally, the development of extrahepatic tumors via chronic damage treatment was investigated.

## 2. Results

### 2.1. Macroscopic and Histological Findings

The animals’ weight was monitored during the 18 weeks of the protocol. At the sacrifice stage, every organ was observed, and macroscopic abnormalities were only found in the liver and lungs of rats in the DEN+2-AAF group. Meanwhile, in the control (CTL) group, there were no signs that suggested the presence of tumors. The livers of the CTL group of rats showed a normal, bright reddish brown coloration with a smooth and shiny surface. The four lobes presented clear margins, which were contrary to the rats from the DEN+2-AAF group, whose livers presented hepatomegaly, pale coloration, well-differentiated nodules, and were stiff to the touch (Figure 1). In this same group, nodules and hemorrhagic areas were observed in the lungs, thus suggesting metastasis.

The liver tissues from the CTL group rats stained with H&E showed a normal lobular organization with cords of hepatocytes that radiated from the central vein toward the portal triads. At higher magnifications, polygonal hepatocytes exhibited ample cytoplasm, as well as round central nuclei with their nucleolus and heterochromatin distributed on the periphery of the nuclear membrane; moreover, the hepatocyte cords were separated by sinusoids without collagen. Conversely, the DEN+2-AAF group exhibited a loss of normal hepatocyte architecture (lobular structure disorder), an increase in atypical cells (enlarged hyperchromatic nuclei and nuclear pleomorphism), and increased mitotic indexes. As a whole, these histological changes were characteristic of the nuclear atypia of neoplastic cells derived from HPCs (Figure 2). Masson staining revealed fibrous septa extending between the portal triads, as well as collagen fibers through the hepatic sinusoids, which are the characteristics of liver cirrhosis.

The CTL group’s lung sections stained with H&E presented a normal morphology; the bronchioles were lined with simple cuboidal epithelia with invaginations toward the lumen, and they were surrounded by concentric smooth muscles of medium thickness, accompanied by small-caliber arteries and veins. The lung parenchyma with alveoli were surrounded by thin septa, which included capillaries that were lined with type I and type II pneumocytes (Figure 3). In contrast, the DEN+ 2-AAF group’s lung tissues showed an increase in the thickness of the alveolar wall due to the infiltration of apparent neoplastic cells and due to the infiltration into the mucosa at the bronchial level. Cells with eosinophilic cytoplasm and hyperchromatic nuclei with irregular nuclear borders were seen. Atypical histological and architecture losses in the lung were found. In focal areas, probable bile pigment (brown spots) was observed (Figure 3). In Masson’s trichrome staining of lung tissues, an increase in alveolar septa with collagen deposits was visualized in the DEN+2-AAF group when compared to the CTL group (Figure 3).

At the end of the 18 weeks of treatment, the animals were sacrificed, and their livers were weighed; the liver-to-body weight ratio was also calculated (Figure 4).

As observed in Figure 4A, there was a decrease in the animals’ weight in the DEN+2-AAF group when compared to the CTL group from the third week until the end of treatment. Moreover, there were statistical increases in the liver weight (hepatomegaly) and liver-to-body weight ratio in the DEN+2-AAF group when compared to the CTL group (*p* < 0.05) (Figure 4B). In addition, the DEN+ 2-AAF group survival rate decreased to 62.5% with animals dying between the ninth and the tenth weeks (Figure 5B). The fibrotic score in the liver was determined based on the classification reported by Ghufran et al. in 2021 [13]: 0 = no fibrosis; 1 = mild fibrosis (collagen fibers extend from the portal triad or central vein to the peripheral region); 2 = moderate fibrosis (collagen fibers form a fibrous septum without compartments); 3 = severe fibrosis (thick fibrous septum accompanied by pseudolobule formation); and 4 = cirrhosis (bridging of fibrous septum around multiple adjacent lobes) (Figure 5A).

### 2.2. Serum Biochemistry Assays

As seen in Table 1, the DEN+2-AAF group had a significant increase in total cholesterol, high density lipoprotein cholesterol (HDL-C), and total proteins, as well as higher activities of aspartate aminotransferase (AST), alkaline phosphatase (ALKP), gamma-glutamyltransferase (GGT), and alanine transaminase (ALT) (*p* < 0.05). No significant differences in glucose, urea, creatinine, triglycerides, and very-low-density lipoprotein (VLDL) levels were observed when compared to the CTL group.

### 2.3. Changes in mRNA Expression Induced by DEN+2-AAF Treatment

Gene expression levels from the liver tissues were determined using RT-qPCR. After 18 weeks of administration of DEN and 2-AAF, a significant increase in *TGF-β1* and *COL1α1* expression (*p* < 0.01) was found, with a 6.2-fold change (FC) and 7.9 FC, respectively, when compared to the CTL group. Furthermore, an increasing tendency in the expression of *COL3α1* and *TIMP* levels was observed; however, these increases were not significant. These data are in concordance with the significant fibrosis that was observed via Masson staining (Figure 2 and Figure 5A).

The DEN+2AAF group presented a decrease in the mRNA levels of *ALB* (*p* < 0.05). However, although AFP has been suggested as an HCC marker, there were, interestingly, no differences in *AFP* or *ERK-2* mRNA levels were found (Figure 6). When comparing the *SOD* and *CAT* mRNA levels between the study groups, statistically significant (*p* < 0.05) decreases were found in the DEN+2-AAF group when compared to the CTL group.

There was a slight, but not significant, decrease in *iNOS* expression in the treated group when compared with the CTL group. Furthermore, inflammatory markers were analyzed: an increase in *TNFa* levels with an FC > 1.8 (*p* < 0.05) and *IL6* levels with FC > 5.3 (*p* < 0.05) were detected in the DEN+2AAF group when compared to CTL group, whereas no significant difference in *IL1B* expression was seen when both groups were compared. In the HCC group, *CYP2E1* expression decreased (*p* < 0.05). Interestingly, differences in the expression of apoptotic, proliferation, and angiogenesis markers such as *BAX*/*BCL-2*, *PCNA*/*HGF*, and *HIF1α*/*VEGF-R2*, respectively, were not statistically significant when comparing the study groups; this was also the case for the *P38*, *PPARα*, and *CPT1A* expression. Nonetheless, in the DEN+2-AAF group, there was an upward trend in the expression levels of *P38*, and a marked downward trend for the *PPARα* and *CPT1A* genes (Figure 6).

## 3. Discussion

HCC is one of the neoplasms with the highest worldwide mortality [6]. The establishment of HCC animal models is an incredibly important tool through which to identify cellular and molecular alterations during hepatocarcinogenesis, as well as for developing efficient treatments [4,18,24]. There are studies where the chronic administration of the carcinogen DEN was delivered alone or in combination with 2-AAF to Fischer 344 or Sprague Dawley rats. This generated multiple liver nodules and carcinomas due to a high rate of liver metabolism driven by CYP450 activation in these rat strains [16,18]. Nonetheless, the Wistar rat strain used in this study is classified as a strain of rat that is of intermediate susceptibility to developing HCC [19,25,26,27]. In this study, for the first time, an HCC model was induced in Wistar rats with the chronic administration of DEN and 2-AAF and gene expression changes of relevant HCC-associated genes were studied. Our results demonstrate that HCC developed, with evident histopathologic damage, indicating an alteration in liver function. At the level of mRNA expression, evident changes in fibrosis (*TGFβ1* and *COL1α1*), inflammation (*TNFα* and *IL6*), oxidative stress (*SOD* and *CAT*), and hepatic metabolism genes (*CYP2E1* and *ALB*) were also noted. It was of interest that no significant changes were detected in *AFP* gene expression. Additionally, lung damage was detected, suggesting a metastasis of the induced primary liver cancer.

To maintain the rapid and uncontrolled proliferation of malignant cells, the organism requires a constant supply of energy and macromolecular components. For this reason, cancer cells reprogram their metabolic pathways and show metabolic alterations; for example, there is increased glucose uptake and dysregulation of fatty acid (FA) metabolism. Intracellular FAs are important for tumor cells to have access to lipids to generate new cell membranes; in addition, cancer cells can obtain the necessary energy through FA oxidation in order to increase cell proliferation and survival [28]. The aforementioned data agree with the chemical hepatocarcinogenesis model used in this study with treated rats having higher glucose, total cholesterol, and HDL-C levels when compared to the CTL group (Table 1). Cholesterol is a major component of the cell membrane and plays an important role in the cell structure. It is also essential for cell proliferation in cancer cells, which have a higher proliferation rate. Serum liver enzymes such as ALT, AST, ALKP, and GGT were used for the liver function evaluation; these enzymes are often elevated in patients with liver disease, and therefore may reflect the liver status [29,30]. These four liver enzymes were found to be significantly increased in the DEN+2-AAF group when compared to the CTL group, which reflects the liver damage caused by the chemical agents used for the hepatocarcinogenesis induction (Table 1) as has been reported by several HCC animal-model studies [31,32,33].

Approximately 90% of HCC cases develop from chronic lesions in the liver, which are lesions that progress from chronic inflammation, fibrosis, and cirrhosis to tumor-nodule formation [2,22]. It has been documented that a constant administration of DEN, or the combination of DEN+2-AAF, generates pathological changes related to the generation of fibrosis and cirrhosis [18,34]. TGF-β1 plays a key role in the pathogenesis of fibrosis, cirrhosis, and HCC. This cytokine is particularly important in regulating tumorigenesis, as well as in controlling numerous cellular functions, such as apoptosis, differentiation, proliferation, and extracellular matrix production [35]. When the TGF-β1 gene and protein expression increases in advanced disease stages, malignant progression is promoted by improving cancer cell survival, epithelial–mesenchymal transition (EMT), migration, invasion, and—ultimately—metastasis [36]. The chemical hepatocarcinogenesis model induced with the chronic administration of the DEN+2-AAF used in this study caused elevated liver *TGFβ1* expression (*p* < 0.01) in the treated group when compared to the CTL group. It has been shown that an elevation in *TGF-β1* expression in the tissue of patients with HCC is correlated with a poor prognosis and short survival time [37].

Fibrosis is characterized by the excessive deposition of extracellular matrix (ECM), which is predominantly composed of a type I collagen, such as COL1α1 and COL3α1, and causes the disruption of the normal tissue architecture leading to liver dysfunction [38]. The type I collagen deposit was evidenced via Masson’s trichrome liver staining in the DEN+2-AAF group, where fibrous septa were observed to be distributed from one portal triad to another and where collagen fibers were running through the liver sinusoids, and by the evident elevation in *COL1α1* expression (*p* < 0.01). This indicates that hepatic stellate cells (HSCs) were activated by the exacerbated expression of *TGFβ1*, which was caused by chronic injury. Activated HSCs also produce tissue-specific inhibitory metalloproteinases (TIMPS), which play an important role by inhibiting matrix metalloproteinases (MMPs), leading to an increase in abnormal fibrotic extracellular matrix [39]. In this study, *TIMP1* expression only showed a tendency to increase, altering the balance of ECM secretion and degradation. Furthermore, in this study, the collagen deposit visualized with Masson’s trichrome staining showed an alteration of the normal architecture of the lung in the DEN+2-AAF group.

DEN was applied as a damage-initiating agent since it is biotransformed, mainly via CYP2E1, by the hepatic cytochrome P450 complex [15]. Then, the 2-AAF produced a clonal expansion of the initiated cells, whereby preneoplastic lesions were formed [40]. In addition, 2-AAF functions as a selective growth inhibitor, and is effective in inhibiting normal hepatocyte proliferation [41]. The DEN-induced carcinogenic activity is caused by DNA alkylation generating O6-ethylguanine, which is one of the main adducts associated with DEN carcinogenesis. Therefore, we decided to analyze the expression of *CYP2E1*; interestingly, a significant decrease in *CYP2E1* (*p* < 0.05) in the DEN+2-AAF group was found. Our results contrast with those reported by Ghufran et al. (2021), and Sivalingam et al. (2018) [16,42] who reported an increase in the expression of this gene. Hakkola et al. (2003) demonstrated that the proinflammatory cytokines IL6 and TNFα inhibit CYP2E1 expression through various mechanisms, including the control of HNF-1a function and the regulation of other transcriptional factors that act on the CYP2E1 5’-upstream regulatory region [43]. Another probable explanation could be the supersaturation of CYP2E1 due to the large amount of ROS generated in the animal model and/or due to the promoter-level mutations or epigenetic changes; however, more studies are required to confirm these theories. In addition, Nishiyama Y. et al. (2016) demonstrated that Wistar rats had the lowest hepatic mRNA expression, protein expression, and enzymatic activity of CYP1A1, 2B1, and 3A2 in comparison to Sprague Dawley, Dark Agouti, and Brown Norway rats, respectively. These results could also most likely be similar in relation to CYP2E1 [25]. Although DEN is metabolized by CYP2E1, as mentioned above, there are reports where *CYP2E1* is downregulated in HCC tissues. This is a regulation that has been associated with the formation of large tumor sizes, poor differentiation, and poor survival [44,45]. As per these previous studies, the *CYP2E1* gene expression decrease found in our study agrees with the formation of large tumor sizes, the severity of liver cancer, and a lower survival rate (62.5%) when compared to the CTL group.

In addition, DEN metabolism induces ROS increase in hepatocytes [15,40]. Due to the importance of ROS in cellular diseases, an antioxidant system for maintaining redox homeostasis in the liver environment is activated. The enzymes CAT and SOD are powerful antioxidants that play a key role in ROS elimination in the cell [46]. A loss of CAT activity during cancer development is associated with tumor formation and metastasis [47]. Prolonged exposure to ROS has been shown to induce the methylation of CpG islands at the CAT promoter in HCC cell lines, affecting its gene expression at the transcriptional level. CpG islands hypermethylation has also been observed in tumor tissues, along with decreased levels of CAT mRNA and protein expression when compared to non-tumor tissues [47]. The chemical hepatocarcinogenesis model used in this study downregulated the *CAT* and *SOD* expression in the DEN+2-AAF group when compared to the CTL group. It is possible that the formation of hydrogen peroxide and superoxide anions was due to antioxidant deficiency.

SOD is a family of enzymes that catalyzes the conversion of the free radical superoxide anion (O_2_^−^) into hydrogen peroxide (H_2_O_2_) and molecular oxygen O_2_; it plays an important role in the antioxidant defense against oxidative stress in cells. However, there are also reports of decreased levels of SOD mRNA and protein, caused by high levels of ROS, in HCC patients. Low levels of SOD expression have been related to the presence of multiple tumors and advanced tumor stages [46]. In our chemical hepatocarcinogenesis model, as previously mentioned, a decrease in *SOD* gene expression levels was identified in the DEN+2-AAF group when compared to the CTL group. Multiple tumors were found in the liver tissues from the treated group, which is consistent with the study of Wang et al. (2016) [46]. In addition, there are reports that demonstrate a decrease in SOD activity, through the acute administration of DEN and 2-AAF, in animal models of hepatocarcinogenesis [48,49]. As such, the reduction in SOD expression found in our study likely depletes the antioxidant capacity, and this effect is caused by the chronic administration of the carcinogens. Although iNOS is an oxidative stress biomarker and its increase has been related to a greater HCC aggressiveness in humans [50], we only observed a downward trend in liver tissues of the DEN+2-AAF group. A significant increase in iNOS expression has already been reported in studies where DEN was administered [16,51]. In addition, NO reacts with free radicals by forming reactive nitrogen oxide species and promotes pathophysiological reactions, such as nitrosation, oxidation, or nitration. The lack of significant changes in iNOS expression may be explained by an increase in TNFα secretion, which has been reported to stimulate gene expression and activity of the key enzyme for regulation of tetrahydrobiopterin (BH4) biosynthesis. This rate-limiting cofactor increases endothelial nitric oxide synthase (eNOS)-derived nitric oxide bioavailability in liver cirrhosis [52].

A drop in apoptosis is related to the development of HCC in addition to the high cell proliferation. There are reports where there is a decrease in the expression of BAX and an increase in the expression of BCL-2 [16,31,32,42,53]. These are accompanied by an increase in the expression of HGF and PCNA, indicating an increase in hepatocyte proliferation [16,33,54]. In this sense, in the DEN+2-AAF group, we only observed a downward trend in BAX expression with no changes in *BCL-2* and upward trends in *HGF* and *PCNA* expression when compared to the CTL group. These results that were found in our study could be due to the type of rat strain used and the chronic 2-AAF administration. Moreover, a statistically significant change with respect to the *HIF1α* gene expression was not found between the groups. Nevertheless, a downward trend in *VEGF-R2* gene expression was observed in this group. There are reports where a high expression of VEGF-R2 in patients with HCC has been correlated with a high AFP serum level [55]. However, it was previously reported that VEGF-R2 was detected in 42% of patients with HCC and this positivity was also correlated with a high serum AFP level. The downward *VEGF-R2* trend found in our study correlated with the absence of significant *AFP* gene expression changes in the treated group, as was similarly observed in patients with HCC [56].

Certain studies have associated a downregulation of FA oxidation with HCC. Key rate-limiting enzymes in lipid catabolism, such as CPT1, are involved in cancer progression [28]. CPT1 is located in the outer mitochondrial membrane and facilitates the transport of long-chain FAs into the mitochondria for β-oxidation by converting them from acyl-CoA to acyl-carnitine. In the liver, CPT1A is the primary isoform expressed, while CPT1B and CPT1C are mainly distributed in muscles, the heart, and the brain [57]. P38 activation induces PPARα downregulation; this, in turn, leads to CPT1A downregulation, without a change in ERK1/2 expression, in the tissues of patients with HCC. These low levels of mRNA in both genes (*PPARα* and *CPT1*) lead to the suppression of FA oxidation, thus resulting in exacerbated cell growth (which has been related to the HCC aggressiveness [58]). In our results, we observed that there was no significant difference in the *ERK-2* expression levels between the CTL and DEN+2-AAF groups; nevertheless, there was a downward trend in the *PPARα* and *CPT1A* levels in the treated group when compared to the CTL group, as well as no significant increase in *P38* expression in the DEN+2-AAF group. According to Li et al. (2015) [58], the aggressiveness of the hepatocarcinogenesis caused in this animal model could be associated with reduced FA oxidation.

The chronic administration of DEN+2-AAF generated sustained inflammation that encouraged fibrosis progression in the development of HCC. IL6, IL1B, and TNFα cytokines were deregulated during chronic inflammation. The activation of signaling pathways associated with inflammation produced ROS and NOS, which exacerbated the malignant microenvironment of the cells and favored hepatocarcinogenesis development [59]. After this analysis of the cytokines’ expressions was conducted in this study, a significant increase in the expression of *TNFα* (*p* < 0.05) and *IL6* (*p* < 0.01), as has been previously reported, was found in the DEN+2-AAF group [16,48,51]. On the other hand, *IL1B* expression was not significantly different between the groups.

AFP has been classified as an HCC marker due to its usefulness in HCC prognosis and diagnosis without biopsy intervention; nonetheless, due to the heterogeneity of this type of cancer, not all HCC tumors are AFP positive [60]. In our study, the determination of *AFP* expression in the liver tissues did not generate a significant difference between both groups. ALB downregulation has been a relevant marker for liver damage, including HCC [61], and ALB downregulation in the tumor tissues of HCC patients has been observed in other studies at both the mRNA and protein levels [62]. ALB expression was significantly different between the groups, which evidenced liver damage in the treated group.

## 4. Materials and Methods

### 4.1. Hepatocellular Carcinoma Model Chemical Induction

The animals were obtained from the Centro de Investigación Biomédica de Occidente (CIBO), Instituto Mexicano del Seguro Social (IMSS). The protocol was approved by the Ethic, Research, and Biosecurity Committee from the “Centro Universitario de Ciencias de la Salud, Universidad de Guadalajara”, approval number: CI-01720. The animals were treated in accordance with the NOM-062-ZOO-1999 guidelines. A total of 13 Wistar rats (180–200 g body weight) were maintained in constant 12 h light/dark cycles and temperature-controlled conditions (22–25 °C). They were fed with a standard rat diet and water ad libitum. Wistar rats were randomly divided into a CTL group and a treated group (DEN+2-AAF). The animals from the experimental group were treated with DEN (Sigma-Aldrich, Inc, St. Louis, MO, USA, 68178) (50 mg/kg/week) intraperitoneally (i.p.). Then, three days after DEN administration, an intragastric dose of 2-AAF (Sigma-Aldrich, Inc, St. Louis, MO, USA, 68178) (25 mg/kg/week) was administered. This scheme was followed over the 18-week experimental period (Figure 7). At the end of the protocol, the rats were anesthetized before blood sample collection (which was performed via cardiac puncture). Livers and lungs were excised and washed in saline solution; then, the liver and lung sections were fixed in 4% formaldehyde for histological analyses. Liver tissue was stored at −80 °C for RNA extraction.

### 4.2. Clinical Biochemistry

Blood samples were centrifuged at 3500 rpm for 10 min and to separate the serum. Glucose, urea, creatinine, total cholesterol, triglycerides, HDL-C, VLDL, total protein, AST, ALKP, GGT, and ALT (Orto Clinical Diagnostics) were quantified from the serum samples by dry chemistry using VITROS 350^®^ Analyzer (Ortho-Clinical Diagnostics, Rochester, NY, USA).

### 4.3. Histological Analyses

Liver and lung tissues were fixed using 4% formaldehyde and then embedded in paraffin blocks. Five-micrometer sections of liver and lung tissues were cut and stained with hematoxylin and eosin as well as Masson trichrome staining, according to the standard procedures. Images were captured and analyzed using a bright-field microscope (Carl Zeiss, Primo Star, Gottingen, Germany), and the histopathological description was performed by two independent pathologists blinded to the treatment groups.

### 4.4. Total RNA Extraction and cDNA Synthesis

RNA extraction was performed using 500 µL of Trizol (Invitrogen, Carlsbad, CA, USA) per 100 mg of tissue. The protocol for the extraction was conducted as per the manufacturer’s instructions. RNA integrity and concentration were determined at a wavelength of 260/280/230 with a NanoDrop^TM^ One^C^ spectrophotometer (Thermo Scientific, Waltham, MA, USA). The RNA was stored at −80 °C until use. The total RNA was reverse transcribed, according to manufacturer’s instructions, to cDNA with an M-MLV RT enzyme (Invitrogen, 200 U/μL).

### 4.5. Gene Expression Analysis

rt-qPCR was performed using LightCycler 96 (Roche, Mannheim, Germany) equipment and the Fast Start Universal SYBR Green Master (Roche; Mannheim, Germany), following the suppliers’ instructions. The primer sequences and rt-PCR cycle programs are shown in Table 2 and Table 3, respectively. *RPL41* was used as a constitutive gene. The specificity of the amplicons was confirmed with the melting curve analysis software of the LightCycler 96 (Version 1.1.0.1320). The data is shown as the fold change using the 2^−ΔΔCt^ method.

### 4.6. Statistical Analysis

Statistical significance was determined by comparing the data from the DEN+2-AAF group against the CTL group, which was achieved by using the Mann–Whitney U test or *t* test, depending on the data normality. A value of *p* < 0.05 was considered significant.

## 5. Conclusions

Chronic administration of DEN+2-AAF induces HCC, which is characterized by fibrogenesis, inflammation, and failure of the antioxidant and hepatic systems. Contrary to the previous reported studies performed in animal models with these chemical carcinogens, liver mRNA expression of *AFP*, as well as apoptotic-, angiogenesis-, and lipid-metabolism-related genes, did not present a significant difference between the treated group and the CTL group. These results support the different signatures of HCC at the molecular level, which could be influenced by the animals’ genetic background and the time–dose route of the administered chemical carcinogens. Despite Wistar rats being considered a strain that possesses an intermediate susceptibility to the development of HCC, the chronic administration of these chemicals over the course of 18 weeks induced HCC and also possible lung metastasis. Future research using this HCC animal model could contribute to a deeper knowledge on cancer progression, thus leading to studies at the genome, transcriptome, proteome, metabolome, and epigenome levels to help us better understand the molecular basis of this cancer. These studies could be very useful in crossing the bridge toward the translational medicine required to improve treatments for HCC and/or to prevent metastasis.

## Figures and Tables

**Figure 1 ijms-24-08387-f001:**
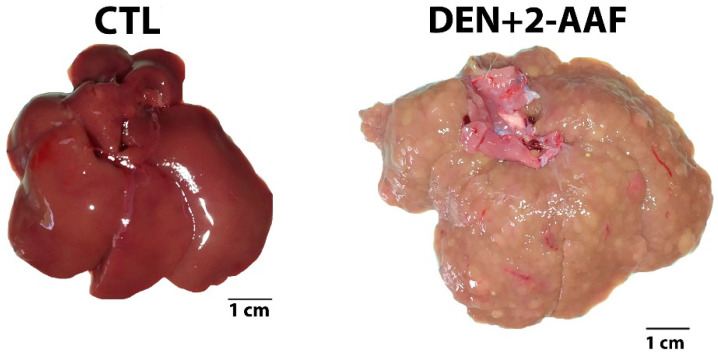
Chemical induction of hepatocellular carcinoma in Wistar rats by chronic administration of DEN and 2-AAF for 18 weeks. CTL: control group without treatment. DEN+2-AAF: treated group administered DEN i.p. 50 mg/kg and 2-AAF orally at 25 mg/kg. Representative liver images.

**Figure 2 ijms-24-08387-f002:**
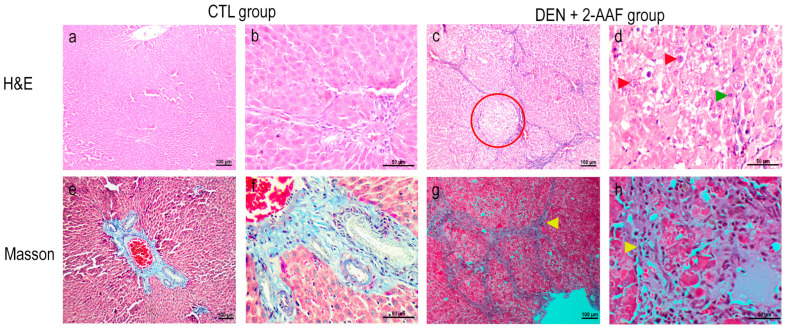
Representative images of the liver tissues stained by Masson’s trichrome and hematoxylin and eosin for both groups. Yellow arrowhead: collagen deposit. Red circle: well-differentiated nodule. Red arrowhead: nuclear pleomorphism. Green arrowhead: atypical mitosis. Images (**a**,**c**,**e**,**g**) were captured using a 10× objective. Images (**b**,**d**,**f**,**h**) were captured with a 40× objective.

**Figure 3 ijms-24-08387-f003:**
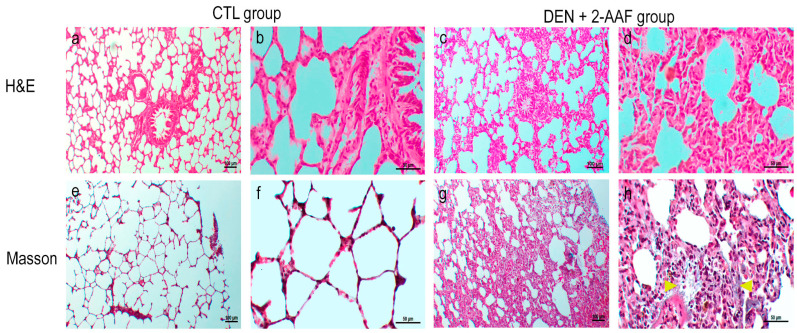
Representative images of the lung tissues stained by Masson’s trichrome and hematoxylin and eosin for both groups. Yellow arrowhead: collagen deposit. Images (**a**,**c**,**e**,**g**) were captured using a 10× objective. Images (**b**,**d**,**f**,**h**) were captured with a 40× objective.

**Figure 4 ijms-24-08387-f004:**
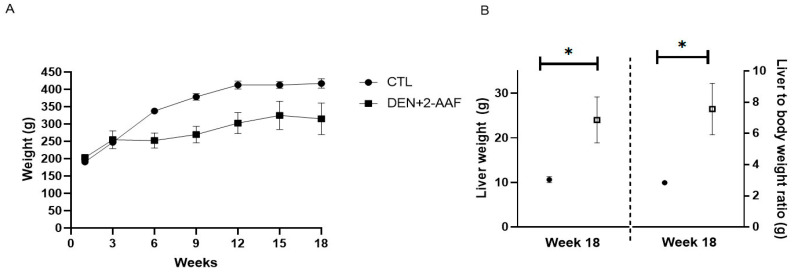
Chronic administration of DEN+2-AAF generates hepatomegaly and a decrease in body weight. (**A**) Body weight throughout the 18 weeks of treatment. Data are represented as the mean ± SD (CTL group, *n* = 5; DEN+2-AAF group, *n* = 5). (**B**) Liver weight and liver-to-body weight ratio of the experimental groups at the end of the 18 weeks of treatment. Data are represented as the mean ± SEM (CTL group, *n* = 5; DEN+2-AAF group, *n* = 5). * *p* < 0.05.

**Figure 5 ijms-24-08387-f005:**
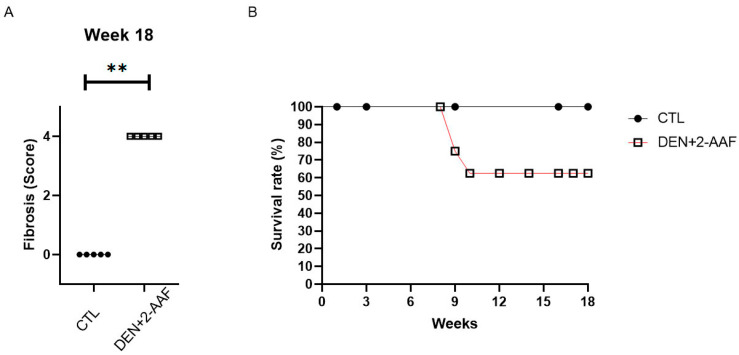
Chemical induction of HCC increases the fibrosis score and reduces the survival rate. (**A**) Histological fibrosis scores in the CTL and DEN+2-AAF groups. Data are represented as the mean ± SD (CTL group, *n* = 5; DEN+2-AAF group, *n* = 8). ** *p* < 0.01. (**B**) Survival rate of both study groups in the hepatocarcinogenesis model. The DEN+2-AAF group presented a survival rate of 62.5% when compared to the 100% survival of the CTL group.

**Figure 6 ijms-24-08387-f006:**
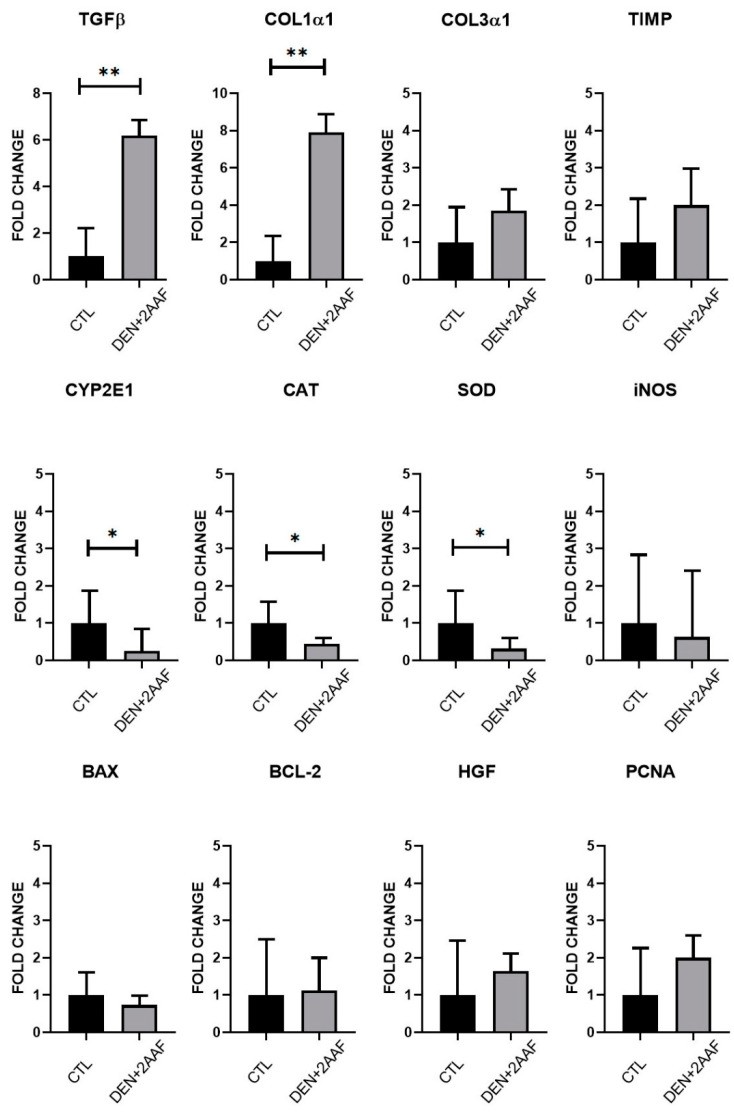
mRNA expression levels in liver tissues. The rt-qPCR results for *TGFβ1* (** *p* < 0.01); *COL1α1* (** *p* < 0.01); *COL3α1*, *AFP*, and *ALB* (* *p* < 0.05); *ERK-2* and *SOD* (* *p* < 0.05); *CAT* (* *p* < 0.05); *iNOS* and *TNFα* (* *p* < 0.05); *TIMP* and *CYP2E1* (* *p* < 0.05); and *BAX*, *BCL-2*, *PCNA*, *HGF*, *HIF1α*, *VEGF-R2*, *P38*, *PPARα*, *CPT1A*, *IL1B*, and *IL6* (** *p* < 0.01) in the CTL and DEN+2-AAF groups. Data are expressed as fold change (2^−ΔΔCt^) based on the ΔCt values compared to the CTL group. Data were normalized to the endogenous *RPL41* gene.

**Figure 7 ijms-24-08387-f007:**
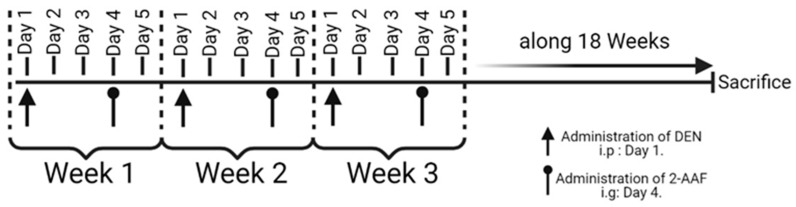
Treatment scheme for chemical induction of HCC. Intraperitoneal injection of DEN (50 mg/kg) and intragastric 2-AAF (25 mg/kg) were administrated weekly during 18 weeks on day 1 and day 4, respectively. After this treatment, the rats were sacrificed. The figure was created with BioRender.com. Agreement number: HV25C1CS7T.

**Table 1 ijms-24-08387-t001:** Glucose, urea, creatinine, total protein, lipid, and hepatic profiles in the male Wistar rat serum from the CTL and DEN+2-AAF groups.

Parameter	CTL Group	DEN+2-AAF Group
Glucose (mg/dL)	153.5 ± 14.34	165.8 ± 13.5
Urea (mg/dL)	44.98 ± 1.88	39.24 ± 6.9
Creatinine (mg/dL)	0.53 ± 0.02	0.58 ± 0.05
Cholesterol (mg/dL)	48 ± 2.62	98.4 ± 13.8 *
Triglycerides (mg/dL)	51.83 ± 7.60	62 ± 8.34
HDL-C (mg/dL)	27.5 ± 2.14	57.8 ± 7.39 *
VLDL (mg/dL)	10.5 ± 1.47	12.2 ± 1.68
Total Protein (g/dL)	6.22 ± 0.19	6.82 ± 0.1 *
AST (U/L)	141.83 ± 11.89	263.8 ± 45.4 *
ALKP (U/L)	171.17 ± 21.39	429.4 ± 30.37 *
GGT (U/L)	5.17 ± 0.16	71.8 ± 14.11 *
ALT (U/L)	35.83 ± 2.31	130.2 ± 22.4 *

Values are presented as the mean ± SE. * *p* < 0.05 compared to the CTL group.

**Table 2 ijms-24-08387-t002:** The primer sequences used for rt-qPCR.

Name	Forward (F) Sequence	Reverse (R) Sequence
P38	F: 5’- AGCAACCTCGCTGTGAATGA	R: 5’- TCCCCGTCAGACGCATTATC
AFP	F: 5’- CTTGGTGAAGCAAAAGCCTGAA	R: 5’- GGACCCTCTTCTGTGAAACAGACT
ALB	F: 5’- CCCGATTACTCCGTGT	R: 5’- TGGCGTTTTGGAATCCATA
BAX **	F: 5′- TGGAGATGAACTGGACAATA	R: 5′- CAAAGTAGAAGAGGGCAAC
BCL2 **	F: 5′- CGACTTTGCAGAGATGTCC	R: 5′- ATGCCGGTTCAGGTACTCAG
CAT	F: 5’- GGAGGCGGGAACCCAATAG	R: 5′- GTGTGCCATCTCGTCAGTGAA
Col1α1 **	S: 5′- CAAGATGGTGGCCGTTACTAC	R: 5′- AGTACTCTCCGCTCTTCCAG
Col3α1 *	S: 5′- TGGACAGATGCTGGTGCTGAG	R: 5′- GAAGGCCAGCTGTACATCAAGG
Cyp2e1 *	S: 5′- CTTTCCCTCTTCCCATCCTT	R: 5′- CCCGTCCAGAAAACTCATTC
CPT1A	F: 5’- GGTCGGAAGCCCATGTTGTA	R: 5’- TTTGGGTCCGAGGTTGACAG
ERK-2	F: 5’- GGAACACTGCATCTTTGAGTGAG	R: 5’- GCACACAGTGCAGGAACAAAA
HGF *	F: 5′- ATGAGAGAGGCGAGGAGAAAC	R: 5′- GTAGCCCCAGCCGTAAATACT
HIF-1α *	F: 5′- CCCATCCATGTGACCATGAG	R: 5′- AATCAGCACCAAGCACGTCA
IL-1β **	F: 5′- CCAAGCACCTTCTTTTCCTTC	R: 5′- GTCAGACAGCACGAGGCATT
IL-6 **	F: 5′- CCACCCACAACAGACCAGTA	R: 5′- CTCCAGAAGACCAGAGCAGAT
iNOS **	F: 5′- GCCCCTTCAATGGTTGGTAC	R: 5′- AGGCCAGTGTGTGGGTCTC
PCNA **	F: 5′- GTGAACCTCACCAGCATGTC	R: 5′- GTTGCTCAACGTCTAAGTCC
PPARα	F: 5′- GTGGTCCCTAATCAGGCCTATATC	R: 5′- ACAATACTACCTGACCACCACT
RPL41	F: 5′- GGCAGAGGTCCAAGTAAACCA	R: 5′- ATCTCGGCGAGGTGACATTC
SOD	F: 5′- AATGTGTCCATTGAAGATCGTGTGA	R: 5′- GCTTCCAGCATTTCCAGTCTTTGTA
TGFβ1	F: 5’- TGCTAATGGTGGACCGCAA	R: 5’- CACTGCTTCCCGAATGTCTGA
TIMP-1 *	F: 5′- TCCCCAGAAATCATCGAGAC	R: 5′- TCAGATTATGCCAGGGAACC
TNF-α **	F: 5′- TGCCTCAGCCTCTTCTCATT	R: 5′- GCTTGGTGGTTTGCTACGAC
VEGF-R2 *	F: 5′- AGATGACAGCCAGACAGACAG	R: 5′- CCACAGACTCCCTGCTTTTAC

Primers with (*) and (**) were reported and modified from Ghufran et al. (2021) [13]. The rest of the primers were designed for this study.

**Table 3 ijms-24-08387-t003:** The rt-qPCR programs for gene amplification.

Number of Cycles	Step	*ALB*, *BCL2*, *CAT*, *COL1α1*, *COL3α1*, *CYP2E1*, *ERK2*, *HGF*, *HIF-1α*, *IL-1β*, *IL-6*, *iNOS*, *PCNA*, *RPL41*, *TGFβ1*, *TIMP-1*, *TNF-α*, and *VEGF-R2*	rt-qPCR Program *CPT1A, P38*, and *SOD*	*AFP*, *BAX*, and *PPARα*
1	Initial Denaturation	95 °C–10 min	95 °C–10 min	95 °C–10 min
40	Denaturation	95 °C–15 s	95 °C–15 s	95 °C–15 s
	Annealing	60 °C–30 s	59 °C–30 s	58 °C–30 s
	Extension	72 °C–25 s	72 °C–25 s	72 °C–25 s

## Data Availability

The raw data are available upon request; please contact the corresponding authors.

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
