# Peer review of "Chronic Administration of Diethylnitrosamine and 2-Acetylaminofluorene Induces Hepatocellular Carcinoma in Wistar Rats"

_ijms, 2023, doi:10.3390/ijms24098387_

Round 1

Reviewer 1 Report

The study presents valuable insights into identifying effective treatments for HCC. However, there are some areas that require improvements.

Firstly, the study lacks a clear research question or hypothesis, which should have been included in the introduction and discussion. I recommend that you consider including a research question or hypothesis in the introduction to clarify the study's purpose and objectives. In addition, providing more background on the available treatments for HCC and the challenges involved in treating this cancer would help the reader understand the need for further research in this area.

Secondly, there are several areas that require improvement in the manuscript.

Line38-39, it would be helpful to introduce the topic more clearly by providing a brief definition or explanation of what HCC is before providing statistics.

Line39-40, the source of the statistics should be included.

Line46-47, it would be beneficial to briefly explain how NAFLD associated with diabetes mellitus or metabolic syndrome has become a more frequent risk factor for HCC development.

Line49-52, the sentence should be rephrased to improve clarity.

Line59-62, it would be helpful to explain why the use of 2-AAF in experimental models for the study of DNA adduct structure, DNA repair, carcinogenesis, and mutagenesis is relevant.

Line82-86, the introduction should explain why HCC heterogeneity is shown in humans and how it correlates with survival.

Finally, the legend titles for Figure 4 and Figure 5 are missing, and I recommend that you add them to the figures.

Best regards

Author Response

Reviewer Comments to Author:

Reviewer: 1

Comments and Suggestions for Authors:

The study presents valuable insights into identifying effective treatments for HCC. However, there are some areas that require improvements.

Firstly, the study lacks a clear research question or hypothesis, which should have been included in the introduction and discussion. I recommend that you consider including a research question or hypothesis in the introduction to clarify the study's purpose and objectives. In addition, providing more background on the available treatments for HCC and the challenges involved in treating this cancer would help the reader understand the need for further research in this area.

Author response & action taken:

R: Thank you very much for your suggestions which were very useful for the manuscript improvement. The manuscript was sent to a professional English editing service. A certificate is added to confirm this information. About your first observation, a research question and treatment were added (lines 135-140).

Secondly, there are several areas that require improvement in the manuscript.

Line38-39, it would be helpful to introduce the topic more clearly by providing a brief definition or explanation of what HCC is before providing statistics.

Author response & action taken:

R: A brief definition of HCC was added (lines 39 to 45).

Line39-40, the source of the statistics should be included.

Author response & action taken:

R: The reference was added (lines 44 to 46).

Line 46-47, it would be beneficial to briefly explain how NAFLD associated with diabetes mellitus or metabolic syndrome has become a more frequent risk factor for HCC development.

Author response & action taken:

R: The explanation was improved to express the association of NAFLD with the HCC development (lines 50 to 55).

Line 49-52, the sentence should be rephrased to improve clarity.

Author response & action taken:

R: The phrase was improved to be more understandable (lines: 58 to 63).

Line59-62, it would be helpful to explain why the use of 2-AAF in experimental models for the study of DNA adduct structure, DNA repair, carcinogenesis, and mutagenesis is relevant.

Author response & action taken:

R: The text was rearranged and more information about 2-AAF was added (lines 69-to 91).

Line82-86, the introduction should explain why HCC heterogeneity is shown in humans and how it correlates with survival.

Author response & action taken:

R: The paragraph was modified adding more information about heterogeneity in HCC (lines 92 to 101).

Finally, the legend titles for Figure 4 and Figure 5 are missing, and I recommend that you add them to the figures.

Author response & action taken:

R: Titles of Figures 4 and 5 were added.

Reviewer 2 Report

Comments listed in the attachment! 

Author Response

Reviewer Comments to Author:

Reviewer: 2

Comments and Suggestions for Authors:

The authors have presented a comprehensive study on the effect of these chemical agents on the development of HCC in a rat model. However, the main objective of this study is not clearly defined. The authors must define why are they investigating these chemical agents and what is the gap in the knowledge in this particular field as doing so will help in determining the novelty and importance of this study.

Author response & action taken:

R: Thank you very much for your suggestions which were very useful for the manuscript improvement. The manuscript was sent to a professional English editing service. A certificate is added to confirm this information. About your first observation, the main objective of this study and the research question was added (lines 135 to 140).

I would also like to suggest the following revisions:

  1. Several data presented in the text are missing appropriate references:
    1. Lines 49-51: “If public health campaigns promoting healthy life styles do not impact and most people continues with obesogenic diets, HCC will be one of the most common causes of death 50 caused by cancer in a near future.”

Author response & action taken:

R: The phrase about public health campaigns… was written by us. Nevertheless, there are reviews suggesting the relevance of obesity and its complications related to HCC (references added) and we found it important to highlight this information since obesity is currently a big problem in Mexico and HCC incidence could spread in the population where obesity is a significant problem.

  1. Lines 54-57: There are strong evidences about the development of HCC in experimental animal models induced by DEN as the only agent or combined with tumor promoting substances like phenobarbital, carbon tetrachloride, ethanol or a high fat diet.

Author response & action taken:

R: This paragraph was rephrased and the information comes from reference 10 as indicated (lines 64 to 69).

  1. Lines 63-67: “DEN and 2-AAF administrated on animal models are bio transformed in the liver by the cytochrome P450 enzymatic system mainly by CYP2E1 (except guinea pigs). As a consequence, generated metabolites interact with DNA to form N7-methylguanine, O6 -ethyl guanine and N-(Deoxyguanosin-8-yl)-2-acetylaminofluor which are the main DNA adducts associated with the carcinogenic power of DEN and 2-AAF, respectively”

Author response & action taken:

R: The text was rephrased and the 4 references are at the end of the paragraph (lines 76 to 83).

  1. Lines 73-76: “2-AAF acts as a tumor promoter, this carcinogenic chemical stimulates DEN-initiated cell proliferation and induces the formation of preneoplastic foci and hyperplastic nodules in the liver. 2-AAF blocks undamaged hepatocytes proliferation due to its mitogen inhibitory activity.”

Author response & action taken:

R: The paragraph was rephrased and the references come at the end of the paragraph (lines 86 to 91).

  1. Lines 63-67: It will be good to include a figure or diagram to better explain the reaction.

Author response & action taken:

R: Even though this information is an important part of the background, a figure was not included since it is not part of the main research.

  1. Lines 71-77: The writing is very complex, may include this part in the figure as well to explain better.

Author response & action taken:

R: The paragraph was arranged for clarity (lines 84-91).

  1. The authors have used several abbreviations: such as CTL, AFP, WNT etc. They must expand these terms at least once in the text

Author response & action taken:

R: Abbreviations were expanded as requested.

  1. Figure 2 has H&E samples as HyE. Also, Figure 2e is missing, the figures are labeled as a, b ,c, d, f, g, h and i.

Author response & action taken:

R: Figures were corrected.

  1. In the discussion section lines 336-338, the authors state that their finding that CYP2E1 is downregulated upon treatment with DEN and 2-AAF, is in contrast to the findings of a previous study. What are the possible reasons for this difference observed.

Author response & action taken:

R: The possible reasons for this difference observed were added (lines 325 to 337).

It will be good to show that protein levels of different genes tested in qRT-PCR analysis as well by means of ELISA or western blot. Reduction in mRNA levels don’t always correlate with reduction in protein levels.

Author response & action taken:

R: We agree with your suggestion, the results could improve considerably with this methodology; nevertheless, this is the first study performed to explore if the chronic administration of DEN and 2-AAF are able to generate an HCC in an intermediate susceptible rat strain (Wistar rat) and to analyze if the gene expression in relevant genes related to fibrosis, inflammation, apoptosis, cell growth, angiogenesis, lipid metabolism, and alpha-fetoprotein (AFP) is modified. The chemical model of HCC induced with both compounds (DEN and 2-AFF) has the great advantage that it is not necessary to perform hepatic hepatectomy in order to develop the HCC after the treatment. The main question was answered, with the biochemical analysis (cholesterol, HDL-C, AST, ALT, ALKP, and GGT) histological findings corroborated with some gene expression changes such as TGFb, COL1a1, CAT, SOD, IL6, TNF-a, and ALB. HCC was generated with abnormalities in liver function, fibrosis induced by changes in TGFb, COL1a1, with altered inflammation IL6, TNF-a and, oxidative stress changes CAT, SOD. On the other hand, it is important to quantify AFP protein in serum, unfortunately, there were not enough serum samples from all animals. Furthermore, we continue at present with studies using this animal model and we are sure that in the near future, more interesting results are going to be obtained that could contribute to the knowledge of this pathology. Additionally, it is relevant that researchers in this area could receive this information that can be applied in their own research areas.

Reviewer 3 Report

Authors present the biochemical, histological, and genetic alterations in rat model of hepatocarcinogenesis administered diethylnitros-20 amine (DEN) and 2-acetylaminofluorene (2-AAF). Overall, the work is well planned, and executed. Comments follow.

MAJOR

1.       The text requires an extensive, detailed English language proofing. I have highlighted some points below.

2.       Discussion: Discussion is best started with a statement  about the uniqueness of the study and the findings.

Also, the discussion is poorly structured and while the argument in itself is strong, the storyline is all over the place.

3.       Line 374-375: The reason for increased iNOS should be explored further in terms of the paradox of nitric oxide in liver disease (DOI: 10.1053/jhep.2002.31432).

MINOR

i.                     Line 23: Please spell out "wk" and "ip".

ii.                   Line 27: "...relates to..."?  or please clarify the sentence.

iii.                 Line 28: spell out AFP.

iv.                 Line 31: "...chronic administration of..." please mention the drug administered..."

v.                   Line 59: Please check and clarify this statement "...[8]. 2-acetylaminoflourene (2-AAF) -developed as an insecticide..."

vi.                 Line 64: "...except in guinea pigs..."

vii.                Lines 73-75: the sentence about 2-AAF is not clear.

viii.              Lines 75-76: sentence about 2-AAF blocking 'UNDERMAGED' do you mean 'normal'?

ix.                  Lines 77-78: Please clarify this sentence "...proliferation is favored inducing hepatic tumors..."

x.                   Line 82: catabolism is a form of metabolism along with anabolism. Please check sentence and rephrase.

xi.                  Lines 83-84: Please rephrase this sentence for clarity "...HCC heterogeneity is shown in human, as an example, differences correlated with high 83 AFP expression are harboring a MYC activation signature ..."

xii.                Lines 87-90: Please summarize or break this long sentence for clarity "....Development of different HCC experimental protocols is a valuable tool to under-87 stand the diverse mechanisms related to HCC induction, progression and metastasis to try to extrapolate with the genotypic and phenotypic variability of this pathology in humans; ..."

xiii.              Line 93: "...In this study, partial..."

xiv.              Lines 94-95: "...in order to accelerate the..."

xv.                Line 97: please insert the ref (no 15).

xvi.              Lines 99-100: remove this line "...which have an intermediate suscepti-99 bility for HCC development, in ..."

xvii.             Line 99: "...for 18 weeks..."

xviii.        Line 101-104: Please remove parenthesis and detail the markers in the method section.

Also, since AFP is appearing for the first time in text, kindly spell out.

xix.              Lines 106-111: Please remove parts about  the summary of findings as this should be moved to the discussion section.

xx.                Line 114: "...during 18 weeks of the protocol..."

xxi.              Line 117: "...there were no changes..."

xxii.             Line 122: Please add CTL in parenthesis next to the "control" mentioned in previous sentences..."

xxiii.           Lines 145-146: Please find alternative to "opposite to,"

xxiv.           Line 213: "...on serum from..."

xxv.             Line 258: "...also a decrease in iNOS expression in treatment group compared with..."

xxvi.           Line 271: "...tool to identify..."

xxvii.         Line286: Cells are not born.

xxviii.        Line 289-290: "...study with treated rats having higher glucose..."

xxix.           Line 310: used "in" this study.

xxx.             All manuscript: Please use treated/treatment group instead of damage group.

xxxi.           Line 315: Please spell out ECM.

xxxii.         Lines 321-322: This statement is not clear "...This indicates that hepatic stellate cells (HSCs) were activated by TGFβ exacerbated expression caused by the chronic injury. ..."

xxxiii.        Line 347: Please spell out SOD (superoxide dismutase) and CAT (catalase).

xxxiv.        Line 357: "...generated by antioxidant deficiency..."

xxxv.          Lines 369-370: Since SOD is an antioxidant, then it is most likely that the downregulation of SOD depletes antioxidant capacity and not the other way round. Please check and correct.

xxxvi.        Line 395: This statement is unclear "Some studies have associated a downregulation of FA oxidation in HCC, the expression levels of most genes related to FA oxidation vary greatly among patients;  "

xxxvii.      Line 409: "...and an increase in p38..."

xxxviii.    Line 409-411: According to which data? please clarify and reference?

xxxix.        Line 417: Analyzes or analysis?

xl.                  Lines 428: groups...damage in this (which?) group?

xli.                Line 440: Spell out IP.

xlii.              Line 477: Trizol for 100mg of tissue?

Author Response

Reviewer Comments to Author:

Reviewer: 3

Comments and Suggestions for Authors

Authors present the biochemical, histological, and genetic alterations in rat model of hepatocarcinogenesis administered diethylnitros-20 amine (DEN) and 2-acetylaminofluorene (2-AAF). Overall, the work is well planned, and executed. Comments follow.

MAJOR

  1. The text requires an extensive, detailed English language proofing. I have highlighted some points below.

Author response & action taken:

R: Thank you very much for your suggestions which were very useful for the manuscript improvement. The manuscript was sent to a professional English editing service. A certificate is added to confirm this information.

  1. Discussion: Discussion is best started with a statement  about the uniqueness of the study and the findings.

Also, the discussion is poorly structured and while the argument in itself is strong, the storyline is all over the place.

Author response & action taken:

R: The first paragraph of the discussion was modified emphasizing the uniqueness of the study and findings (lines 247 to 263). The end of the first paragraph about CYP1A1, 2B1, and 3A2 mRNA explanation was modified and relocated to the paragraph related to CYPE21 discussion results (lines 328-330).

  1. Line 374-375: The reason for increased iNOS should be explored further in terms of the paradox of nitric oxide in liver disease (DOI: 10.1053/jhep.2002.31432).

Author response & action taken:

R: The eNOS paradox was included as well as the specified reference (lines 369-373).

MINOR

  1. Line 23: Please spell out "wk" and "ip".

Author response & action taken:

R: Line 23 was modified. wk and ip were spelled out.

  1. Line 27: "...relates to..."?  or please clarify the sentence.

Author response & action taken:

R: The phrase corresponding to line 27 was modified to make the information clear (lines 28 o 29).

iii.                 Line 28: spell out AFP.

Author response & action taken:

R: AFP was spelled out.

  1. Line 31: "...chronic administration of..." please mention the drug administered..."

Author response & action taken:

R: Drugs were mentioned (line 32)

  1. Line 59: Please check and clarify this statement "...[8]. 2-acetylaminoflourene (2-AAF) -developed as an insecticide..."

Author response & action taken:

R: The statement about -acetylaminoflourene (2-AAF) -developed as an insecticide was modified (lines 69-75).

  1. Line 64: "...except in guinea pigs..."

Author response & action taken:

R: The paragraph was modified (line 76-83).

vii.                Lines 73-75: the sentence about 2-AAF is not clear.

Author response & action taken:

R: The sentence about 2-AAF was modified (lines 86-91).

viii.              Lines 75-76: sentence about 2-AAF blocking 'UNDERMAGED' do you mean 'normal'?

Author response & action taken:

R: UNDERMAGED was not fund, probably you refer to undamaged, anyway the paragraph was rephrased to clarify (Lines 86-91).

  1. Lines 77-78: Please clarify this sentence "...proliferation is favored inducing hepatic tumors..."

Author response & action taken:

R: It is part of the previous modified paragraph (lines 86-91).

  1. Line 82: catabolism is a form of metabolism along with anabolism. Please check sentence and rephrase.

Author response & action taken:

R: The word catabolism was changed and the full paragraph was rephrased (lines 92-98).

  1. Lines 83-84: Please rephrase this sentence for clarity "...HCC heterogeneity is shown in human, as an example, differences correlated with high 83 AFP expression are harboring a MYC activation signature ..."

Author response & action taken:

R: Both lines were modified when rephrasing the statement (lines 98-102).

xii.                Lines 87-90: Please summarize or break this long sentence for clarity "....Development of different HCC experimental protocols is a valuable tool to under-87 stand the diverse mechanisms related to HCC induction, progression and metastasis to try to extrapolate with the genotypic and phenotypic variability of this pathology in humans; ..."

Author response & action taken:

R: The statement was modified (lines 123-128).

xiii.              Line 93: "...In this study, partial..."

Author response & action taken:

R: The word mentioned was modified (line 130).

xiv.              Lines 94-95: "...in order to accelerate the..."

Author response & action taken:

R: Lines 94 -95 were modified (lines 132-133).

  1. Line 97: please insert the ref (no 15).

Author response & action taken:

R: Reference was added with the specified reference with the new number. (line 135).

xvi.              Lines 99-100: remove this line "...which have an intermediate suscepti-99 bility for HCC development, in ..."

Author response & action taken:

R: The phrase was removed and relocated in parenthesis in line 137.

xvii.             Line 99: "...for 18 weeks..."

Author response & action taken:

R: The complete phrase was rewritten (lines 142-143).

xviii.        Line 101-104: Please remove parenthesis and detail the markers in the method section.

Also, since AFP is appearing for the first time in text, kindly spell out.

Author response & action taken:

R: Parenthesis were written according to the Journal instructions and all the abbreviations were spelled out (lines 142-158).

xix.              Lines 106-111: Please remove parts about  the summary of findings as this should be moved to the discussion section.

Author response & action taken:

R: Lines 106-111 were moved to discussion section (lines 257-263).

  1. Line 114: "...during 18 weeks of the protocol..."

Author response & action taken:

R: The phrase was modified (line 161).

xxi.              Line 117: "...there were no changes..."

Author response & action taken:

R: The phrase was modified (line 164).

xxii.             Line 122: Please add CTL in parenthesis next to the "control" mentioned in previous sentences..."

Author response & action taken:

R: Control (CTL) was specified (line 163).

xxiii.           Lines 145-146: Please find alternative to "opposite to,"

Author response & action taken:

R: Opposite to was changed by In contrast (line 188).

xxiv.           Line 213: "...on serum from..."

Author response & action taken:

R: The paragraph was removed since it was part of the methodology (between lines 210-211).

xxv.             Line 258: "...also a decrease in iNOS expression in treatment group compared with..."

Author response & action taken:

R: The line was rephrased (line 234-235).

xxvi.           Line 271: "...tool to identify..."

Author response & action taken:

R: The line was rephrased (line 248).

xxvii.         Line286: Cells are not born.

Author response & action taken:

R: The phrase was corrected (lines 268-269).

xxviii.        Line 289-290: "...study with treated rats having higher glucose..."

Author response & action taken:

R: Lines were corrected as suggested (lines 271–272).

xxix.           Line 310: used "in" this study.

Author response & action taken:

R: Preposition was changed (line 293).

xxx.             All manuscript: Please use treated/treatment group instead of damage group.

Author response & action taken:

R: “Damage group” was modified in all cases.

xxxi.           Line 315: Please spell out ECM.

Author response & action taken:

R: “ECM” was spelled out (line 297).

xxxii.         Lines 321-322: This statement is not clear "...This indicates that hepatic stellate cells (HSCs) were activated by TGFβ exacerbated expression caused by the chronic injury. ..."

Author response & action taken:

R: The statement was rephrased (lines 303-305).

xxxiii.        Line 347: Please spell out SOD (superoxide dismutase) and CAT (catalase).

Author response & action taken:

R: + SOD and CAT were spelled out in the introduction (line 148).

xxxiv.        Line 357: "...generated by antioxidant deficiency..."

Author response & action taken:

R: The word was changed as suggested (line 350).

xxxv.          Lines 369-370: Since SOD is an antioxidant, then it is most likely that the downregulation of SOD depletes antioxidant capacity and not the other way round. Please check and correct.

Author response & action taken:

R: Lines 369-370 were rephrased.

xxxvi.        Line 395: This statement is unclear "Some studies have associated a downregulation of FA oxidation in HCC, the expression levels of most genes related to FA oxidation vary greatly among patients;  "

Author response & action taken:

R: The phrase was corrected (lines 362-363).

xxxvii.      Line 409: "...and an increase in p38..."

Author response & action taken:

R: The phrase was corrected (line 403).

xxxviii.    Line 409-411: According to which data? please clarify and reference?

Author response & action taken:

R: Reference was included (line 404).

xxxix.        Line 417: Analyzes or analysis?

Author response & action taken:

R: Line was corrected  (line 411).

  1. Lines 428: groups...damage in this (which?) group?

Author response & action taken:

R: Line 428 was corrected.

xli.                Line 440: Spell out IP.

Author response & action taken:

R: ip was spelled out (lines 434-435).

xlii.              Line 477: Trizol for 100mg of tissue?

Author response & action taken:

R: Trizol was used per 100 mg of tissue (line 455-456) .

Round 2

Reviewer 2 Report

I am satisfied with the author's response to my queries. 

Reviewer 3 Report

I congratulate the authors for this impressive work and wish them best of luck.